# Scale-Up of Biosynthesis Process of Bacterial Nanocellulose

**DOI:** 10.3390/polym13121920

**Published:** 2021-06-09

**Authors:** Nadezhda A. Shavyrkina, Vera V. Budaeva, Ekaterina A. Skiba, Galina F. Mironova, Nikolay V. Bychin, Yulia A. Gismatulina, Ekaterina I. Kashcheyeva, Anastasia E. Sitnikova, Aleksei I. Shilov, Pavel S. Kuznetsov, Gennady V. Sakovich

**Affiliations:** 1Laboratory of Bioconversion, Institute for Problems of Chemical and Energetic Technologies, Siberian Branch of the Russian Academy of Sciences (IPCET SB RAS), 659322 Biysk, Altai Krai, Russia; 32nadina@mail.ru (N.A.S.); eas08988@mail.ru (E.A.S.); yur_galina@mail.ru (G.F.M.); nbych@yandex.ru (N.V.B.); julia.gismatulina@mail.ru (Y.A.G.); makarova@ipcet.ru (E.I.K.); sitnikova97.97@mail.ru (A.E.S.); shilov_36@mail.ru (A.I.S.); black_heart98@mail.ru (P.S.K.); admin@ipcet.ru (G.V.S.); 2Biysk Technological Institute, Polzunov Altai State Technical University, 659305 Biysk, Altai Krai, Russia

**Keywords:** bacterial nanocellulose, biosynthesis scale-up, *Medusomyces gisevii*, BNC characterization

## Abstract

Bacterial nanocellulose (BNC) is a unique product of microbiological synthesis, having a lot of applications among which the most important is biomedicine. Objective complexities in scaling up the biosynthesis of BNC are associated with the nature of microbial producers for which BNC is not the target metabolite, therefore biosynthesis lasts long, with the BNC yield being small. Thus, the BNC scale-up problem has not yet been overcome. Here we performed biosynthesis of three scaled sheets of BNC (each having a surface area of 29,400 cm^2^, a container volume of 441 L, and a nutrient medium volume of 260 L and characterized them. The static biosynthesis of BNC in a semisynthetic nutrient medium was scaled up using the *Medusomyces gisevii* Sa-12 symbiotic culture. The experiment was run in duplicate. The BNC pellicle was removed once from the nutrient medium in the first experiment and twice in the second experiment, in which case the inoculum and glucose were not additionally added to the medium. The resultant BNC sheets were characterized by scanning electron microscopy, capillary viscosimetry, infrared spectroscopy, thermomechanical and thermogravimetric analyses. When the nutrient medium was scaled up from 0.1 to 260 L, the elastic modulus of BNC samples increased tenfold and the degree of polymerization 2.5-fold. Besides, we demonstrated that scaled BNC sheets could be removed at least twice from one volume of the nutrient medium, with the yield and quality of BNC remaining the same. Consequently, the world’s largest BNC sheets 210 cm long and 140 cm wide, having a surface area of 29,400 cm^2^ each (weighing 16.24 to 17.04 kg), have been obtained in which an adult with burns or vast wounds can easily be wrapped. The resultant sheets exhibit a typical architecture of cellulosic fibers that form a spatial 3D structure which refers to individual and extremely important characteristics of BNC. Here we thus demonstrated the scale-up of biosynthesis of BNC with improved properties, and this result was achieved by using the symbiotic culture.

## 1. Introduction

Worldwide interest in bacterial nanocellulose (BNC) continues unabated and is due to its properties such as a 3D porous structure with nanosized fibers, high purity, high crystallinity and water-holding capacity, considerable mechanical strength and elasticity, high degree of polymerization, and excellent biocompatibility [1]. It is because of these properties that BNC has found a wide spectrum of applications, ranging from the food industry to medicine. BNC can be also used either as a component of food products or as a food packaging material [2]. But BNC is demanded most of all in medicine to improve health and save human lives. BNC is used as artificial skin, vascular grafts, dental implants, artificial bones and cartilages, a sorbent for targeted delivery of drugs, proteins and hormones, as well as for sorption and pinpoint use of stem cells [3].

While investigating properties of BNC, such characteristics were discovered that make BNC an irreplaceable material in wound healing. The water-holding capacity of BNC and the nanostructured morphology of its fibers similar to the extracellular matrix protein, i.e., collagen, make BNC a highly promising material for cell immobilization [4,5]. These facilitate faster regenerative processes in wound healing [6,7]. The high biocompatibility of BNC mitigates risks of rejection and inflammatory response when placing a wound dressing and reduces the risk of fibrotic scarring [8,9]. Besides, it is because of the nanofiber network that BNC creates a physical barrier which prevents the wound site from bacterial infiltration and minimizes the risk of infections [10]. The extremely high water-holding capacity (1 g dry BNC is capable of holding 99 g water) keeps a moist environment around the wound, preventing it from drying out, and fosters outflow of wound exudate [11,12,13]. The biodegradation property of BNC can be exploited to make dressings that will need no removal, that is, no painful feelings when the wound is undressed, which is topical for chronic wounds [14,15].

All those properties allow BNC to be reckoned as a revolutionary wound-healing material, which makes research on BNC production technology highly relevant and important. The key difficulty that restrains adoption of BNC as a massive wound-healing dressing is the issue of its production scale-up. The large pharma Johnson & Johnson attempted to commercialize BC as far back as 1980s, particularly as a dressing to accelerate wound healing, but details of any clinical trials have never been made public and the commercial product has not been produced [16,17]. In the 1990s, another study into the commercial production of BNC was undertaken by a series of large Japanese companies and government-owned organizations for the purpose of massive production of BNC [18]. About $45 billion was expended for this research, resulting in numerous patents and publications, but no successful commercial production was subsequently established. In the 1990s, basic research on BNC biosynthesis was also pursued in Poland, whereby an effective *Gluconacetobacter* strain was discovered that is capable of producing cellulose in more economical nutrient media [19]. Thus, the focus of scientific research has turned towards discovering new microbial producers of BNC that would allow the improvement in productivity of BNC biosynthesis.

A 2016 book by Gama [1] discusses a project of scaled production of bacterial cellulose as a hydrogel, with a capacity of more than 500 tons a year, and outlines economic cost estimations for this production: with a final cost of 1 kg BNC hydrogel at $25, the payback period of investments in production is expected to be 4 years.

The objective complexity in scaling up the BNC biosynthesis process [20] stems from the nature of microbial producers for which BNC is not the target metabolite, therefore biosynthesis takes a long time and the yield of BNC is not high: the biosynthesis time for static culture is 5 to 20 days, with the yield not exceeding 8 g/L [21]. Nonetheless, the high value of BNC justifies great costs for its production.

Works on enhancing the BNC yield include the following directions: a search for and genetic construction of new high-productivity microbial producers [1,5,21,22]; design of new technological approaches to cultivation [5,20,21,23]; and a quest for alternative and cheap raw sources [5,24,25,26,27,28,29]. Sharma [5] stresses that the high capital investment together with high operating costs is the major constraint towards commercialization of BNC at a reasonable cost.

Among the promising directions in boosting the productivity of the BNC production process is upgrading fermentation equipment and, in this case, a question arises about the way how to organize cultivation, whether it be stationary or dynamic, because microbial producers of BNC are obligate aerobes? Choi et al. [30] reported a pilot-scale production under agitated conditions in a 50-L modified bubble column bioreactor of airlift type using saccharified food waste as the feedstock. Stirred-tank bioreactors serve as a tool to scale up production and lead to higher production yields of BNC [31]. However, the index of crystallinity of such a cellulose is as low as 37%. Besides, cellulose obtained under stirred conditions has limitations in use.

At the same time, agitation does not allow homogeneous, quality BNC films to be obtained. This results in a fibrous or granulated material [32]. The yield of the resultant BNC is usually lower than that in static culture because the shear stress promotes mutation of bacteria into strains that do not produce cellulose [33]. Kralisch et al. [34] reported a 180-L reactor with horizontal lift (HoLiR). It combines both strategies, the static and agitated culture, and provides semi-continuous cultivation and harvesting of flat non-woven materials and BNC films. Those authors note that the semi-continuous biosynthesis technique and process scale-up decrease production costs appreciably.

Proceeding from the set objective of producing a bacterial cellulose sheet with such dimensions and high strength behavior so that an adult having vast burns could be wrapped in it, we elected to perform a scaled cultivation by the static method that gives access to an unbroken structure of cellulosic fibers.

In the present study, we performed biosynthesis of three scaled sheets of BNC (each having a surface area of 29,400 cm^2^, a container volume of 441 L, and a nutrient medium volume of 260 L) and investigated their properties. To the best of our knowledge, this is the largest-scale stationary cultivation in the world practice.

## 2. Materials and Methods

### 2.1. Biosynthesis of BNC

#### 2.1.1. Microbial Producer of BNC

The *Medusomyces gisevii* Sa-12 symbiotic culture acquired from the All-Russian National Collection of Industrial Microorganisms (Scientific Center ‘Kurchatov Institute’, Research Institute for Genetics and Selection of Industrial Microorganisms, Moscow, Russia) was used herein as the microbial producer. The producer has advantages of being tolerant to the foreign microflora and stably producing BNC. The required amount of the inoculum (26 L) was prepared from the lab-grown culture in a pilot-scale fermentor. For the inoculum preparation, we used a semisynthetic nutrient medium consisting of glucose (20 g/L) and black tea extract (5 g/L) [35]; the culture purity was monitored by microscopy.

#### 2.1.2. Preparation of Nutrient Medium (260 L) for Scaled Cultivation

A 234-L nutrient medium was prepared in a stainless-steel stirred reactor at the pilot production site of the IPCET SB RAS. For scaled cultivation, we used a semisynthetic medium with the same composition as that used for the inoculum preparation: 20 L/g glucose (CAS No. 219 5996-10-1, OOO PromSintez, Chapayevsk city, Russia) and 5 g/L black tea extract. Ready-to-use water was poured into the reactor and heated to boil; weighed portions of glucose and black tea were added to the boiling water, and the reactor heating was tuned off. The medium was held for 30 min, then filtered and put into a biosynthesis container.

The biosynthesis container was a stainless-steel vessel of 210 cm in length, 140 cm in width and 15 cm in height, with a total volume of 441 L. On scaled cultivation, the container was filled with the nutrient medium to 60% of its volume, more specifically to 260 L. The container had these dimensions because the idea was to produce a BNC sheet having such a size so that, if necessary, an adult could be wrapped in it.

#### 2.1.3. Biosynthesis of BNC Sheet

Biosynthesis of BNC sheets was performed subject to the following conditions: no agitation and temperature and humidity in the room were maintained automatically at 27 °C and 85%, which is optimum for the microbial producer used. After being filled with the water and inoculum, the container was covered with a sterile non-woven material (Megaspan Agro light-stabilized covering fabric, Fuleren Factory, Barnaul city, Russia), which provided oxygen supply to the microbial producer and protected the container from potential contaminants during biosynthesis.

Biosynthesis of large-scale BNC sheets was run in the following manner. The first cultivation was continued for 14 days, then the sheet (designated as Sheet 1) was taken out of the culture medium surface and washed, and the remaining nutrient medium was not used further. In the second experiment, the container was filled with a nutrient medium, the inoculum was added, the biosynthesis was continued under the aforesaid conditions for 14 days, and then the sheet was taken out (designated as Sheet 2/1) and washed, and the remaining nutrient medium was left for further biosynthesis for another 14 days; a fresh inoculum and a fresh portion of glucose were not injected. After a 14-day cultivation, one more sheet (designated as Sheet 2/2) was taken out and washed as well.

The concentration of glucose in the culture medium was analyzed by a UNICO UV-2804 spectrophotometer (United Products & Instruments Inc., Dayton, NJ, USA) using 3,5-dinitrosalycilic acid (Panreac, Barcelona, Spain) as reagent [36]. The active acidity of the medium was measured potentiometrically by an I-160MI ion meter (OOO Izmeritelnaya Tekhnika, Moscow, Russia).

#### 2.1.4. Removal and Washing of BNC Sheet

After the biosynthesis process was complete, the sheets were taken out of the growth medium surface and washed; the washing routine is detailed elsewhere [35]. Figure 1 displays photographs of the BNC sheet when it was in the biosynthesis container and when removed from the nutrient medium surface.

BNC was washed at room temperature by successively immersing the BNC sheets into a 15-fold volume of a 2% NaOH solution (CAS No. 1310-73-2, OAO Kaustik, Volgograd, Russia), then into distilled water (until the gel-films turned white), next into a 0.1% HCl solution (CAS No. 7647-01-0, AO LenReaktiv, Saint-Petersburg, Russia), and finally into distilled water (until neutral wash waters). As the BNC sheets had a heavy weight and a great thickness, the washing took long and required that wash waters be replaced by fresh reagent solutions 5–6 times in each case. Photographs of the BNC sheets before and after washing are shown in Figure 2.

After the washing was complete and the sheets were held on a kapron sieve until the water completely ran off, the sheets were weighed on a platform electronic balance (ZAO MASSA-K, Saint-Petersburg, Russia).

#### 2.1.5. Moisture Measurement of BNC Sheets

To measure the moisture content, samples of the BC pellicles washed off of cell debris and weighing 1.0 g to an accuracy of 0.002 g were dried at 120 °C to constant weight and then weighed again. The moisture was estimated as the difference in weight of the sample before and after drying [37].

#### 2.1.6. Calculation of BNC Yield

The yield of BNC (%) was calculated by Equation (1):(1)$$\eta =\frac{m_{BNC}}{C_{g}\cdot V\cdot 0.9}\cdot 100$$
where *η* is the BNC yield, %; *m_BNC_* is the BNC sample weight on an oven-dry basis (g); *C**_g_* is the glucose concentration in the medium (g/L); *V* is the nutrient medium volume (L); and 0.9 is the conversion factor attributed to the water molecule detachment when glucose is polymerized into cellulose. The yield was calculated as reported [38].

In calculating the yield of the second BNC sheet taken out of the same nutrient medium (designated as Sheet 2/2 in the experiment), we used the same formula but took into account the altered parameters such as glucose concentration and nutrient medium volume because some of the medium (including glucose) had been consumed for biosynthesis of the first sheet designated as Sheet 2/1.

#### 2.1.7. Structural Characterization of BNC

The surface morphology of BNC fibers was studied by scanning electron microscopy (SEM) on a JSM-840 scanning electron microscope (JEOL Ltd., Tokyo, Japan) after sputtering a Pt layer 1–5 nm thick. The BNC sample having dimensions of 5 cm × 5 cm was preliminarily held in aqueous ethanol solutions at different concentrations of 25%, 50%, 75% and 90% for 30 min in four steps for partial dehydration of the BNC sample; afterwards, the BNC sample was freeze-dried in an HR7000-M lyophilizer (Harvest Right, LLC, Salt Lake City, UT, USA).

Scanning electron microscopy (SEM) of freeze-dried BNC samples pre-dehydrated with ethanol was done using a JSM-840 microscope (JEOL Ltd., Tokyo, Japan) with a Link-860 series II X-ray microanalyzer. Repeats information given above

#### 2.1.8. Measurement of Degree of Polymerization

The BNC sample required for analysis was preliminarily freeze-dried.The degree of polymerization of BNC samples was measured by the viscometric method [39] using cadoxene as solvent (ethylenediamine, CAS No. 107-15-3, AO LenReaktiv; cadmium oxide, CAS No. 1306-19-0, AO LenReaktiv). The capillary viscosimetry was performed on a VPZh-3 viscometer with a capillary diameter of 0.92 mm (Ekroskhim, Saint-Petersburg, Russia).

#### 2.1.9. Infrared Spectroscopy

The BNC sample required for analysis was preliminarily freeze-dried. Infrared spectroscopy was performed on an Infralum FT-801 FTIR spectrophotometer (OOO NPF Lumex Sibir, Novosibirsk, Russia) at 4000–500 cm^−1^ in KBr pellets.

#### 2.1.10. Thermomechanical Analysis of BNC Samples

The BNC sample required for analysis was preliminarily freeze-dried. The strength of BNC was measured on a DTG-60 thermomechanical analyzer (Shimadzu, Kyoto, Japan) whereby the test sample was stretched at a rate of 5.0 g/min from 0.0 g to a maximum load of 400.0 g until failure; the test temperature was 23.0 °C.

The thickness of BNC samples was measured by a ICh-10 I-class dial indicator thickness gauge (Kirov factory ‘Krasny Instrumentalshchik’, Kirov city, Russia). Thermomechanical analysis of the samples to measure the strength behavior was performed on a Shimadzu TMA-60 thermomechanical instrument. The test samples were stretched at a rate of 5.0 g/min to a maximum load of 500 g at room temperature.

The elastic modulus was estimated by Equation (2):(2)$$E=\frac{\sigma_{sx}}{\left(\frac{\varepsilon_{sx}}{100}\right)}$$
where *E* is the modulus of longitudinal elasticity, MPa; $\sigma_{sx}$ is the conventional yield limit, MPa; $\varepsilon_{sx}$ is the relative elongation at yield, %.

#### 2.1.11. Thermogravimetric Analysis of BNC Samples

The BNC sample required for analysis was preliminarily freeze-dried. Thermogravimetric analysis was performed on a Shimadzu DTG-60 thermogravimetric analyzer. The test conditions were as follows: the test sample was heated at a rate of 10 °C/min to a maximum temperature of 600 °C in nitrogen environment at a flow rate of 40 mL/min. The sample weight was *p* = 6.0–6.5 mg. The analyses were done using equipment provided by the Biysk Regional Center for Shared Use of Scientific Equipment of the SB RAS (IPCET SB RAS, Biysk, Russia).

## 3. Results and Discussion

Biosynthesis of BNC

The scaled cultivation resulted in three pearl-white BNC sheets. Each sheet had dimensions same as the container, and the surface area of the sheet was 29,400 cm^2^. The sheets were visually even, nicely smooth to touch, and 0.6 to 0.7 cm thick. The basic indicators of the biosynthesis are summarized in Table 1.

Unique findings have been obtained in the course of this study: biosynthesis of BNC has been performed for the first time in a 260-L nutrient medium and BNC sheets have been produced, each weighing 16–17 kg. Besides, we demonstrated that two BNC sheets could be taken out of one volume of the nutrient medium. That is, the following situation was observed: at an initial glucose concentration of 20.1 g/L, the BNC sheet weighed 16.24 kg in 14 days of cultivation, the residual glucose concentration was 5.9 g/L. Upon further cultivation for another 14 days, one more BNC sheet was obtained, weighing 16.58 kg, with the glucose concentration declined to 2 g/L. As was shown previously, a decrease in the initial glucose concentration below 20 g/L adversely influences the BNC biosynthesis process [35]. How then the biosynthesis of the second BNC sheet can be explained if the initial glucose concentration was 5.9 g/L? As is well known, cellulose fibrils are synthesized by microbial producers through the formation of intermediates: glucose 6-phosphate is isomerized to glucose 1-phosphate which reacts with uridine triphosphate (UTP), forming uridine-5′-diphosphate-*α*-D-glucose (UDP-glucose). It is this compound that is attacked by the enzyme (cellulose synthase) that is responsible for transfer of glycosylic residues from UDF-glucose to the growing chain of *β*-D-1,4-glucan [22]. Apparently, the intermediates required for BNC synthesis are accumulated inside the microbial producer cells during the biosynthesis and gradually opt in the formation process of cellulosic fibers which are released outside from the cells. It is because of the accumulation of initial components of cellulose biosynthesis inside microorganisms that a few BNC sheets can be obtained in one nutrient medium, despite the fact that the amount of reducing sugars in the medium after the first removal of a BNC pellicle diminishes very considerably.

The second explanation of the observed phenomenon of the synthesis of the second BNC sheet in the nutrient medium with a low glucose concentration is that the *Medusomyces gisevii* symbiotic culture develops its own metabolic pathway during its evolution, in which all the nutrients required for the normal functioning of the symbiosis are synthesized by the symbiotic culture itself. This culture is capable of switching from one substrate to another and utilizing its own metabolic products as substrates. The culture is known to be capable of utilizing a number of organic acids such as acetic, gluconic, succinic, lactic, malic and glycerol, which are its metabolic products [40].

Compared to the previous findings on biosynthesis of BNC in a 72-L nutrient medium [38] for 7 days of cultivation, the yield declined by 1.4 times (from 5.20% to 3.45–3.69%) in case the sheet was taken out once, and the yield increased by 1.3 times from 5.20% to 6.98% in case two BNC sheets were removed from one volume of the nutrient medium (260 L). If the yields of the BNC sheets are compared within the given experiment, it can be noted that the overall yield of BNC in the second experiment is almost twice that in the first experiment: 6.98% (Sheet 2/1 + Sheet 2/2) versus 3.69% (Sheet 1). That said, the objective of producing a strong BNC sheet with dimensions allowing an adult to be wrapped therein has been achieved. A photo of an adult wrapped in the BNC sheet is depicted in Figure 3.

Molina-Ramírez et al. [25,26] described a 40-fold scale-up starting from a 90-cm^3^ glass vessel to a plastic container 62.5 cm long, 29 cm wide and 18.5 cm high; the culture medium volume was raised from 45 cm^3^ to 1800 cm^3^. The results were obtained using an alternative nutrient medium, i.e., overripe banana, and a *Komagataeibacter medellinensis* strain. The yield (in total volume, g/L) was 5 in the first case and 4 in the second; the dry weight (g) was 0.46 and 16, respectively. That said, those authors concluded that the quantity of the BNC obtained in the scale-up was 16 g, 3378% greater than that obtained in the glass vessel under optimized conditions. Following this logic, the quantity of the BNC obtained in the scale-up in our case was 16.24–17.04 kg, which is 173,504–182,051% greater than that obtained in the 0.1-L nutrient medium.

Figure 4 illustrates SEM images of the BNC sheets. As can be seen in the images, all three sheets exhibit a representative architecture of cellulosic fibers: each BNC sheet represents a bundle of cellulosic nanofibrils as clusters of stretched cellulosic chains that form a spatial 3D structure which relates to individual and highly important characteristics of BNC [1,41].

Table 2 lists cellulose degrees of polymerization, strength behavior and TGA data for BNC. For the all three BNC sheets, the degree of polymerization had high values ranging from 4900 to 5200. The elastic modulus was also high, between 9180 and 9687 MPa, in all the three cases. It is noteworthy that for the second removed Sheet 2/2, the degree of polymerization was 12% greater and the elastic modulus was 5.5% higher than for the first removed Sheet 2/1, with the weights of the sheets being in fact identical. This is a very optimistic fact for industrial cultivation: two BNC sheets can be taken out of the single nutrient medium without impairment of BNC quality.

By evaluating the mechanical characteristics of the BNC sheets and comparing them with the previous data for the same cultivation conditions, nutrient medium composition and the microbial producer used [28], it can be noted that when the biosynthesis process of BNC is scaled up by 2600 times, the elastic modulus rises tenfold (for samples obtained in a 0.1-L nutrient medium, this parameter was 933 MPa), while the degree of polymerization increases 2.5-fold (2200 versus 5500). Thus, when the BNC biosynthesis is scaled up, the strength of the samples and the degree of polymerization of cellulosic fibers increase. These findings are of scientific interest because Molina-Ramírez et al. [31] reported a decrease in elastic modulus from 1149.84 MPa to 569.04 MPa when they scaled up biosynthesis by 44 times from 90 mL to 4 L using an alternative medium—overripe Banana juice—and a *Komagataeibacter medellinensis* strain. In this case, the elastic modulus was noticed to decline twofold, whereas the decomposition temperature and crystallinity index were approximately equal [25,26]. 

The TGA data listed in Table 2 are given for all the three stages representative of BNC [42,43]. The first stage is associated with evaporation of highly volatile components (water) of the sample over a temperature range from 25 °C to 150 °C. At the second stage, the sample undergoes a maximum weight loss, which is due BNC being decomposed by pyrolysis over a temperature range between 150 °C and 400 °C. The third stage shows re-decomposition of the material, which is ascribed to degradation of the polymeric chains and the six-membered cyclic structure of pyran [43], and this stage is observed over a temperature range from 400 °C to 600 °C. The TGA data demonstrate very close values of decomposition temperature of the samples (315.71–316.54 °C;) and the residues after decomposition (24.12–25.27%), indicating that the samples are identical in structure and have a high purity.

Figure 5 depicts an infrared spectrogram of the BNC sheets. The obtained IR spectra of all the three BNC sheets were almost identical and demonstrated a pattern typical of cellulose: absorption bands at 3363 cm^−1^ were attributed to stretch vibrations of OH groups and absorption bands at 2895 cm^−1^ were due to stretch vibrations of CH_2_ and CH groups. Absorptions bands at 1634 cm^−1^ were for bending vibrations of OH groups of strongly bound water. Absorption bands near 1430–1370 cm^−1^ suggested bending vibrations of CH_2_ and CH groups. Absorption bands near 1340–1320 cm^−1^ indicated bending vibrations of the primary OH alcohol group, those near 1290–1240 cm^−1^ stood for bending vibrations of CH_2_ alcohol group, and those near 1160–1040 cm^−1^ referred to stretch vibrations of C–O–C and C–O alcohol groups. Absorption bands at 899 cm^−1^ corroborated the presence of *β*-1,4 glycosidic bonds of glucose [44,45].

## 4. Conclusions

The scale-up of static biosynthesis of BNC in a semisynthetic nutrient medium in a 441-L metal container using the *Medusomyces gisevii* Sa-12 symbiotic culture afforded three strong BNC sheets, each having an area of 29,400 cm^2^ and a weight ranging from 16.24 to 17.04 kg, in which an adult having burns or vast wounds can easily be wrapped. This finding is pioneer among the examples of static culture on account of the maximum nutrient medium volume of 260 L and the high cellulose degree of polymerization and strength of the resultant sheets. When the biosynthesis was scaled up from 0.1 L to 260 L of the nutrient medium, the elastic modulus of the BNC samples increased tenfold and the degree of polymerization by 2.5 times, which is attributed particularly to the symbiotic culture used. Besides, it was demonstrated that at least two scaled BNC sheets could be taken out of one volume of the nutrient medium without reduced yield and quality of BNC.

## Figures and Tables

**Figure 1 polymers-13-01920-f001:**
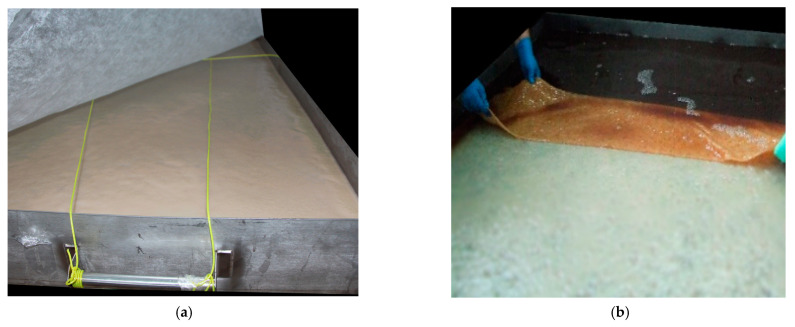
Biosynthesis of BNC sheets: (**a**) BNC sheet in the biosynthesis container 14 days post-cultivation and (**b**) removal of BNC sheet from the nutrient medium surface.

**Figure 2 polymers-13-01920-f002:**
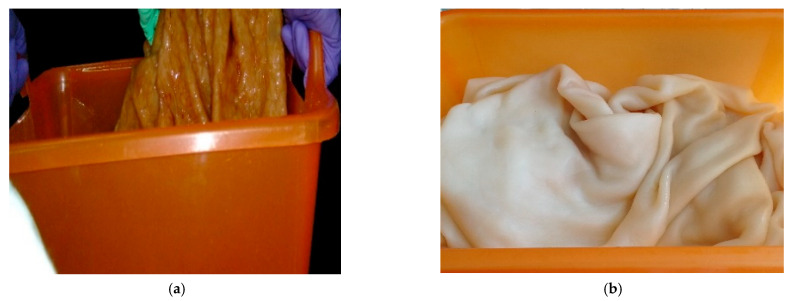
Washing of BNC sheets: (**a**) unwashed BNC sheet and (**b**) washed BNC sheet.

**Figure 3 polymers-13-01920-f003:**
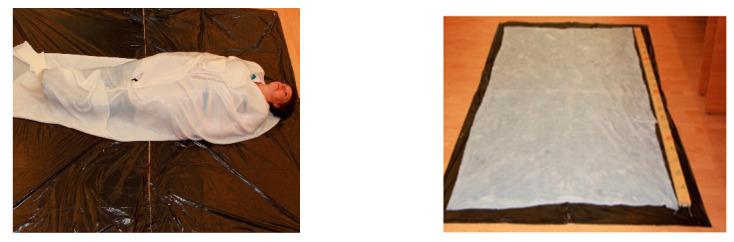
An adult wrapped in the BNC sheet (the ruler is 2 m long).

**Figure 4 polymers-13-01920-f004:**
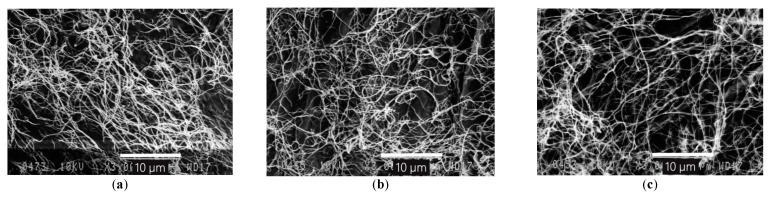
SEM images of BNC: (**a**) Sheet 1, (**b**) Sheet 2/1 and (**c**) Sheet 2/2 at a zoom of ×3000.

**Figure 5 polymers-13-01920-f005:**
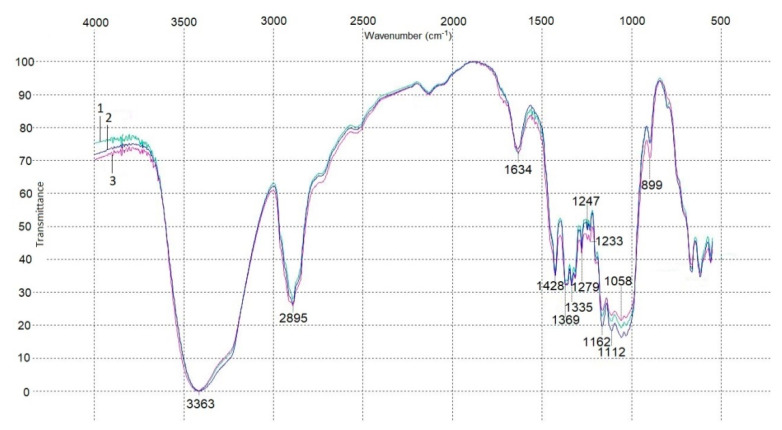
IR spectroscopy data for BNC sheets: 1—Sheet 1; 2—Sheet 2/1; and 3—Sheet 2/2.

**Table 1 polymers-13-01920-t001:** Basic indicators of BNC biosynthesis.

Characteristics	Sheet		
	1	2/1	2/2
Initial volume of nutrient medium (*V*, L)	260	260	229.52
Residual volume of nutrient medium (L)	228.4	229.52	198.30
pH of medium at the onset of biosynthesis	3.8	3.9	2.7
pH of medium upon completion of biosynthesis	2.8	2.7	2.3
Initial glucose concentration in nutrient medium (*C_g_*, g/L)	19.94	20.10	5.90
Residual glucose concentration in nutrient medium (g/L)	6.80	5.90	2.00
Weight of BNC sheet (kg)	17.04	16.24	16.58
Moisture of BNC sheet (%)	98.99	99.01	99.00
BNC yield (*η*, %)	3.69	3.45	13.60
		6.98	

**Table 2 polymers-13-01920-t002:** Cellulose degree of polymerization, strength characteristics and TGA data for BNC.

Characteristic	Sheet		
	1	2/1	2/2
Degree of polymerization	4920	4900	5500
**Strength behavior**			
Conventional yield limit, (σ_sx_, MPa)	54.55	52.31	50.49
Strength at break (σ_r_, MPa)	160.41	155.32	151.12
Relative elongation at break (ε_τ_, %)	2.94	2.71	2.60
Relative elongation at yield (ε_sx_, %)	0.59	0.54	0.55
Elastic modulus (E, MPa)	9246	9180	9687
**Summary TGA data**			
Sample weight variation in the first stage, %	2.56	3.37	2.43
Sample weight variation in the second stage (in a range of sample decomposition), %	62.54	61.25	62.11
Onset temperature of decomposition, °C	316.20	315.71	316.54
Sample weight variation in the third stage, %	10.78	10.11	10.22
Residue, %	24.12	25.27	25.24

## Data Availability

The raw/processed data required to reproduce these findings are available from the author upon request.

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
