# Peer review of "Scale-Up of Biosynthesis Process of Bacterial Nanocellulose"

_polymers, 2021, doi:10.3390/polym13121920_

Round 1
Reviewer 1 Report
Dear colleagues,
The article entitled “Scale-up of biosynthesis process of bacterial nanocellulose” has the main goal of aiding to overcome BNC scale-up problem. Resulting sheets showed potential to dress vast wounds, namely burn-derived wounds. Content is relatively new and interesting, but please comment the works with doi: 10.3390/pr8111469, 10.1016/j.msec.2019.109963, 10.1002/jctb.5699, 10.1016/j.biortech.2017.09.089 and 10.1016/j.bej.2017.07.007. It is well-written, but novelty and applicability could be further justified. Experimental work could also be further examined. Please revise it. Detailed suggestions can be found below:
Abstract:
“Thus, the BNC scale-up problem has not yet been overcome.” – why exactly? Briefly, what has been done in the literature, and how does your findings seem to be more promising? Where exactly is your novelty?
“Consequently, the world’s largest BNC sheets weighing 16.24 to 17.04 kg have been obtained in which an adult having burns or vast wounds can easily be wrapped.” – it is very interesting that the authors have such a clearly defined application in mind, and one that makes sense. Please add dimensions of the sheets in addition to their weight, and succinctly explain the biomaterial and sheet properties (rather than suitable size) that turn it appealing for this application.
Introduction:
Lines 44-46: “BNC is like an extracellular matrix ─ connective tissue of living organism, even though BNC is composed of polymeric glucose units, while the extracellular matrix is made up of glycoproteins” – may resemble the collagenous structure of the ECM. I understand that the authors want to showcase BNC as an attractive biomaterial, but inconsistencies should be avoided. Please study in detail ECM of the skin, including chemical structures of the main connective tissue fibres and ground substance. Use concrete details to describe its main appealing features for skin regeneration. It is still unclear to me why should we look at BNC in this way rather than other promising biomaterials. This is important to highlight in your manuscript.
Lines 88-89: “biosynthesis takes a long time and the yield of BNC is not high.” - Please add concrete values of biosynthesis time and BNC yield, additionally indicating which values would be considered ideal.
Materials and Methods:
Lines 191-192: “A 5 cm Ń… 5 cm BNC sample was preliminary held in ethanol and then freeze-dried.” – I have difficulties in understanding this step. In a potential application, you intend to use freeze-dried sheets? Indeed, that makes sense, particularly concerning sterility and stability under storage. If that is the case, that should be properly justified in the manuscript. But why the ethanol? Is it to turn the mesh more compact so that pores are then smaller after freeze-drying? That step will alter the architecture. It is to be done each time? If so, further conditions need to be added to the text, namely how was it done (immersion, etc.), time of contact and if it was 100% absolute ethanol or diluted in water. If they re-gain hydration, they could be used for smaller areas. However, if freeze-dried, then it would be less flexible, unable to fold as you intended to for the final application. Why not cryo-TEM or cryo-SEM?
Results and Discussion:
3.1 Biosynthesis of BNC – the same conditions should be done at least three times. Two times would be OK if the results on this were complementary to the study, but this is the core of the work performed. You need more. For me, like this it is unacceptable, as it could be due to chance. It would be a pity, because the article has the foundations to be very interesting. Looking at the materials and methods, time of culture and further processing, you would need nearly a month to do this.
For TGA and FTIR, freeze-drying is suitable, but morphology and mechanical properties should be done with the sheet in their correct state: hydrated (as you show in Figure 3) or dried (after freeze-drying). However, in the characterization, you only compare the three sample types. You should add suitable controls.
Conclusions: please rewrite based on the aforementioned comments.
Author Response
The author's response to reviewer 1 has been uploaded as a separate Pdf file.

Reviewer 2 Report
The manuscript entitled "Scale-up of biosynthesis process of bacterial
nanocellulose" reports an interesting study dealing with the biosynthesis of scaled sheets of bacterial nanocellulose and the characterization of the resulting materials.
In my view, the manuscript deserves to be published on Polymers.
Minor remarks:
- Please, add the scale bars in the SEM micrographs reported in Figure 4
- TGA data reported in Table 2 should be clarified. Which is the meaning of first, second and third stage? Which is the degradation mechanism?
Author Response

(The authors gave the same response as above.)

Round 2
Reviewer 1 Report
You have answered to most of my concerns. Good job.
Best of luck for future work.